# Antiphotoaging and Skin-Protective Activities of *Ardisia silvestris* Ethanol Extract in Human Keratinocytes

**DOI:** 10.3390/plants12051167

**Published:** 2023-03-03

**Authors:** Lei Huang, Long You, Nur Aziz, Seung Hui Yu, Jong Sub Lee, Eui Su Choung, Van Dung Luong, Mi-Jeong Jeon, Moonsuk Hur, Sarah Lee, Byoung-Hee Lee, Han Gyung Kim, Jae Youl Cho

**Affiliations:** 1Department of Biocosmetics, Sungkyunkwan University, Suwon 16419, Republic of Korea; 2Department of Integrative Biotechnology, Sungkyunkwan University, Suwon 16419, Republic of Korea; 3Pharmacy Program, Faculty of Science and Technology, Ma Chung University, Malang 65151, Indonesia; 4DanjoungBio, Co., Ltd., Wonju 26303, Republic of Korea; 5Department of Biology, Dalat University, 01 Phu Dong Thien Vuong, Dalat 66106, Vietnam; 6National Institute of Biological Resources, Environmental Research Complex, Incheon 222689, Republic of Korea; 7Research Institute of Biomolecule Control and Biomedical Institute for Convergence at SKKU (BICS), Sungkyunkwan University, Suwon 16419, Republic of Korea

**Keywords:** *Ardisia silvestris* ethanol extract, UVB irradiation, ROS, AP-1, anti-apoptosis, antioxidative capacity

## Abstract

*Ardisia silvestris* is a traditional medicinal herb used in Vietnam and several other countries. However, the skin-protective properties of *A. silvestris* ethanol extract (As-EE) have not been evaluated. Human keratinocytes form the outermost barrier of the skin and are the main target of ultraviolet (UV) radiation. UV exposure causes skin photoaging via the production of reactive oxygen species. Protection from photoaging is thus a key component of dermatological and cosmetic products. In this research, we found that As-EE can prevent UV-induced skin aging and cell death as well as enhance the barrier effect of the skin. First, the radical-scavenging ability of As-EE was checked using DPPH, ABTS, TPC, CUPRAC, and FRAP assays, and a 3-(4-5-dimethylthiazol-2-yl)-2-5-diphenyltetrazolium bromide assay was used to examine cytotoxicity. Reporter gene assays were used to determine the doses that affect skin-barrier-related genes. A luciferase assay was used to identify possible transcription factors. The anti-photoaging mechanism of As-EE was investigated by determining correlated signaling pathways using immunoblotting analyses. As-EE had no harmful effects on HaCaT cells, according to our findings, and As-EE revealed moderate radical-scavenging ability. With high-performance liquid chromatography (HPLC) analysis, rutin was found to be one of the major components. In addition, As-EE enhanced the expression levels of hyaluronic acid synthase-1 and occludin in HaCaT cells. Moreover, As-EE dose-dependently up-regulated the production of occludin and transglutaminase-1 after suppression caused by UVB blocking the activator protein-1 signaling pathway, in particular, the extracellular response kinase and c-Jun N-terminal kinase. Our findings suggest that As-EE may have anti-photoaging effects by regulating mitogen-activated protein kinase, which is good news for the cosmetics and dermatology sectors.

## 1. Introduction

The greatest organ in the human body and the interaction with the outside world is the skin. The skin serves as a barrier that protects against infections, physical and chemical harm, and uncontrolled water loss [1]. It is made up of the epidermis, dermis, and subcutis, three separate layers [2]. Ninety percent of the epidermis, which is the skin’s outermost layer, is made up of keratinocytes [3]. The stratum corneum (SC), which is the epidermis’ outermost layer and the first line of defense, plays a crucial role in maintaining the integrity of the skin barrier as well as the skin’s suppleness [4,5]. Several studies have reported increased hydration of the SC and improved epidermal barrier function, and thus have potential applications in moisturizing, protective, and anti-aging cosmetics [6,7,8].

Skin aging is a multifactorial process that causes functional and cosmetic changes in the skin. Unlike other organs of the human body, the skin is impacted not only by the intrinsic aging process, but also by several extrinsic environmental elements that accelerate aging, particularly ultraviolet (UV) radiation. The primary environmental element causing early skin aging is UV exposure (photoaging). Based on the amount of sun exposure and the quantity of skin pigment, UV irradiation causes human skin to age over time [9,10]. UV energy can be subdivided into UVA, -B, and -C components based on electrophysical properties, with UVC photons having the shortest wavelengths (100–280 nm) and the greatest amount of energy, UVB falling between the two, and UVA having the longest wavelengths (315–400 nm) but the least-energetic photons. Each UV component has different potential impacts on molecules, cells, and tissues [11]. Because UVB radiation is directly absorbed by DNA and is known to produce cyclobutane pyrimidine dimers and 6-4 pyrimidine pyrimidone dimers, it is predominantly a DNA-damaging agent [12,13,14]. Unrepaired DNA lesions result in DNA mutation during cell division, which may initiate carcinogenesis [15]. UV rays from the sun damage keratinocytes and fibroblasts at the molecular level, activating cell surface receptors that start signal transduction cascades [16]. Solar UV radiation activates a protein from the serine/threonine protein kinase family that is linked to cellular signaling, namely the mitogen-activated protein kinase (MAPK) pathway. In general, the MAPK pathways are divided into three distinct pathways: c-Jun NH2-terminal kinase (JNK), extracellular signal-regulated kinase (ERK), and p38 MAPK (p38 kinase). The ERK cascade promotes cell proliferation and survival, whereas the other two pathways (JNK and p38 kinase) protect and promote apoptosis, respectively [17,18,19]. By targeting distinct intracellular proteins, each member of the serine/threonine protein kinase family generates a different stimulus or cellular stress. ERK activation is normally caused by UVA-mediated reactive oxygen species (ROS), whereas JNK is mostly activated by UVC, and p38 kinases may be activated by all UV wavelengths (including UVA, UVB, and UVC) to modulate DNA damage response [20].

Nearly all eukaryotic cells produce ROS and regulate various physiological processes. However, excessive ROS can cause tissue malfunction and oxidative damage by changing the structural and functional characteristics of cellular constituents such proteins, lipids, and nucleic acids [21,22,23]. Our skin is shielded from harm by the antioxidant defense system, such as UV radiation. ROS, which are brought on by oxidative stress or UV light, accelerate the aging and wrinkle-causing processes in the skin. ROS overproduction initiates internal cellular apoptosis or programmed cell death [24,25,26]. Apoptosis is supposed to accelerate aging or diseases associated with aging. For these reasons, consuming antioxidants to eradicate free radicals is one strategy to maintain healthy skin or stop aging [27,28,29,30]. According to the theory of free radical aging (FRTA) [31], one of the primary causes of aging and diseases associated with aging is the buildup of oxidative stress brought on by ROS [32,33,34]. Even at relatively low concentrations compared to the levels of oxidizable substrates, antioxidants are chemicals that can significantly slow down or block the oxidation of oxidizable substrates [32,35,36]. Although organisms contain defense mechanisms to counteract the oxidative effects of ROS, these mechanisms must be supported by exogenous antioxidants when the balance between ROS formation and antioxidant systems is aberrant [37,38,39].

As the quality of life improves, photoaging has been treated through diet, hormone therapy, and probiotics. However, in recent years, there has been a growing interest in natural herbal cosmetics to combat photoaging, and various botanical extracts have been launched that claim to reduce skin aging and enhance skin health [40,41,42,43,44]. *Ardisia silvestris* was received from the National Institute of Biological Resources (Ministry of Environment, Incheon, Korea). The leaves of As-EE were collected from Vietnam on 30 August 2016. As-EE is a common plant with a wide distribution, especially in Vietnam. However, its properties have yet to be validated scientifically. Until now no studies have shown that As-EE has skin-protective functions. The goal of this study was to assess the anti-aging activity of As-EE and to investigate its skin-protective functions in terms of anti-oxidant capacity, anti-apoptosis, and moisturizing effects under UVB irradiation. This study used cosmetological and pharmacological procedures such as antioxidant assays, mRNA preparation, reverse transcription polymerase chain reaction, immunoblotting analysis, and PI/Annexin V staining with FACS to investigate the potential usefulness of As-EE in skin care. Since we have very promising results on As-EE’s beneficial role in skin, this extract can be applied to the development of cosmeceutical preparations.

## 2. Results

### 2.1. Effects of As-EE on Antioxidative Capacity

The anti-oxidation activity of As-EE was examined using the 2,2-di(4-tert-octylphenyl)-1-picrylhydrazyl (DPPH) assay, 2,2′-Azinobis-(3-ethylbenzothiazoline-6-sulfonic acid (ABTS) assay, cupric ion reducing antioxidant capacity (CUPRAC) assay, and ferric-reducing antioxidant power (FRAP) assay. Its inhibitory concentration of 50% (IC_50_) values for DPPH and ABTS were 46 µg/mL and 13 µg/mL. First, DPPH tests were utilized to assess the ability of natural components to scavenge free radicals [45]. Stable DPPH• free radicals in the DPPH assay lose their purple hue following reduction. At concentrations of 3.125–200 µg/mL, As-EE scavenged DPPH• radicals in a dose-dependent manner, and it began to exhibit considerable DPPH radical-scavenging action at 12.5 µg/mL (Figure 1a). The ABTS assay was examined to ascertain the anti-oxidative-stress effect of As-EE. As potassium persulfate or manganese dioxide oxidize ABTS, they produce bluish-green ABTS•+ radicals that become less pigmented when they are reduced by antioxidants [46]. The ABTS-radical-scavenging activity was inhibited dose-dependently by As-EE at a concentration of 3.125~200 µg/mL. The antiradical activity of As-EE at a concentration of 12.5 µg/mL was similar to that of ascorbic acid used as a positive control (Figure 1b). Similar to the FRAP assay, the CUPRAC assay uses metal ions, except Cu is used in place of Fe. At 450 nm, the hue changes from light blue to orange-yellow when Cu (II) is reduced to Cu (I) by a reducing agent [46]. Cu ions were dose-dependently decreased by As-EE at concentrations from 50–200 µg/mL, which is readable at 450 nm, as shown in Figure 1c; however, the effect was significantly less than that of trolox 0.4 mM. The FRAP assay is based on the idea that antioxidants cause colorless Fe3+-TPTZ to be converted to intensely blue Fe2+-TPTZ, which can be read at 593 nm. Trolox (Sigma, St. Louis, MO, USA) was employed as the antioxidant standard at a total concentration of 0.4 mM [46]. We found that the ferric-reducing antioxidative capacity was reduced by As-EE dose-dependently (Figure 1d). The components of As-EE were analyzed with high-performance liquid chromatography (HPLC) (Figure 1e–m). Rutin, quercetin, hesperidin, and kaempferol were used as standard compounds for HPLC analysis (Figure 1e,h,k). As shown in Figure 1f, rutin and quercetin were detected in As-EE, while hesperidin and kaempferol were not identified. The contents of rutin and quercetin were calculated to be 0.53 and 0.03 mg/g, respectively, using standard area curves of these compounds (data not shown). The conducive pharmacological activities of these components, such as anti-inflammatory and anti-oxidative effects, have been reported previously [47,48,49,50], implying that As-EE has potential in preventing UVB-induced skin damage. We also determined the As-EE’s potential as an antioxidant by checking its total phenolic content (TPC), then we evaluated the As-EE’s antioxidative activity. In terms of gallic acid equivalenst, the TPC is 140 ± 0.02 mg per g of As-EE [51].

### 2.2. Effects of As-EE on Cell Viability and Skin Moisture Protection Activity

Before measuring the skin-protective activity of As-EE in HaCaT cells (human keratinocytes), we evaluated the cytotoxicity of As-EE using the MTT assay. The cell viability results showed that As-EE did not induce the cell death of HaCaT cells (Figure 2a,b). To determine the skin-protective and skin-hydration efficacies of As-EE, the mRNA expression levels of occludin and HAS-3 were detected with reverse transcription-polymerase chain reaction (RT-PCR). As shown in Figure 2c,d, the mRNA expression levels of occludin and HAS-3 were significantly increased by 100 µg/mL of As-EE. To identify the As-EE-dependent signaling pathway, RT-PCR was used to detect the mRNA expression of occludin, transglutaminase (TGM)-1, and HAS-1 in cells treated with inhibitors of MAPK such as SB203580 (a p38 inhibitor), SP600125 (a JNK inhibitor), U0126 (an ERK inhibitor), and Bay117082 (an inhibitor of kappa B kinase (IKK) inhibitor). The mRNA expression of occludin, which was increased by As-EE, was suppressed by SP600125. However, the mRNA expression of TGM-1 and HAS-1 was not reduced by SP600125 in As-EE-treated HaCaT cells (Figure 2e). As-EE increased the expression of epidermal barrier and hydration genes through modulating the expression of occludin via the JNK-dependent signaling pathway.

### 2.3. Effects of As-EE on MAPK-Mediated AP-1 and CREB Signaling Pathway

Occludin is known to be regulated by transcription factors [52]. To investigate the regulator of HAS1 and HAS3, we evaluated the effect of As-EE on the transcriptional activity of CREB (Figure 3a). The results showed that As-EE increased CREB-mediated luciferase activity and the phosphorylation of CREB (Figure 3b). The phosphorylated levels of c-Jun, c-Fos, and JNK were also increased in HaCaT cells treated with As-EE (Figure 3c). Vitamin E (tocopherol) is widely recognized as a potent antioxidant commonly used in topical skin care products [53,54]. Despite its widespread use as a lipid-soluble antioxidant, few studies have examined the moisturizing effects of vitamin E on the skin. We checked the gene expression levels of hydration factors and the skin barrier using RT-PCR with different concentrations of vitamin E. As shown in Figure 3d, we found that mRNA levels of these factors were significantly increased by treatment with 12.5 μM vitamin E. Moreover, CREB-mediated luciferase activity was activated by As-EE in a dose-dependent manner (Figure 3e. In addition, the level of p-JNK was also enhanced in a dose-dependent manner from 0 to 12.5 µM, with the JNK total form invariant (Figure 3f). Thus, As-EE induced the transcriptional factor CREB through the c-Jun, c-Fos, and JNK-dependent signaling pathways.

### 2.4. Moisturizing and Anti-Apoptotic Effect of As-EE in UVB-Irradiated Human Keratinocytes

Previous reports suggest that the ability of UVB radiation to impair the skin immune system has been widely documented, and UVB-induced damage is a key factor in the development of sun-induced skin cancer [55,56]. To explore the potential ability of As-EE in protecting keratinocytes against UVB irradiation, the morphological changes in As-EE-treated HaCaT cells stimulated with UVB were detected using phase-contrast microscopy. Expectedly, the number of floating dead cells was reduced in the As-EE-pre-treated group (Figure 4a). To identify whether the viability of cells is reduced by UVB, cell viability was analyzed with an MTT assay. The cell death that is induced by UVB was inhibited by treatment with As-EE (50~100 µg/mL) (Figure 4b). Apoptosis is a well-known mode of programmed cell death that occurs in multicellular organisms and is used not only to control tissue homeostasis but also to remove severely damaged cells and to protect from the excess growth of abnormal cells in cancer in the epidermis of human skin, which consists mainly of keratinocytes and is constantly renewed. Thus, keratinocyte apoptosis plays a crucial role in the maintenance of epidermal structure and function. However, regulated cell death may be perturbed by environmental factors, particularly UVB, leading to sunburn (keratinocytes undergo UVB-induced apoptosis) and impairing skin integrity. In this study, we suggested the potential of As-EE to modulate UVB-induced apoptosis in human keratinocytes. To investigate the effects of As-EE on cell death in HaCaT cells further, propidium iodide (PI)–annexin V staining and FACS were used. Although UVB radiation caused cell death in HaCaT cells, pre-treatment with As-EE reduced cell death from 58% to 45%, as seen in Figure 4c. To examine whether As-EE also played a role in preserving moisture levels in human keratinocytes under UVB irradiation, the mRNA levels of skin barrier factors were determined using RT-PCR. The results showed that As-EE heightened the expression of occludin and TGM-1 in a dose-dependent manner. Especially when treated with 75 µg/mL and 100 µg/mL of As-EE, the gene expression levels recovered remarkably (Figure 4d,e). Moreover, we investigated the protein levels of UVB-mediated conditions in HaCaT cells using immunoblotting. We found that As-EE down-regulated the phosphorylation of ERK and CREB (Figure 4f–i). This indicated that As-EE repaired cell damage from UVB exposure. Finally, we treated human keratinocyte cells with 20 μM of three inhibitors, SB203580 (p38 inhibitor), SP600125 (JNK inhibitor), and U0126 (ERK inhibitor), related to the AP-1 pathway under UVB irradiation. We found that inhibitors of ERK and JNK notably extricated the expression of occludin and TGM-1, which was inhibited by UVB irradiation (Figure 4j,k). To sum up, these findings indicated that the JNK and ERK signaling pathways predominantly contribute to the moisture-retaining capacity). This implies that As-EE ameliorates the damage to skin after exposure to UVB by increasing the expression of JNK and CREB.

Here, we have identified that As-EE can lead to changes in biomarkers linked to skin hydration and prevent the photodamage of keratinocytes from UVB irradiation by restraining the JNK/ERK/AP-1 and CREB pathways, as well as by inhibiting apoptosis (Figure 4).

## 3. Discussion

The skin not only defends the body from environmental change and pathogenic infection, but it also inhibits moisture loss, allowing homeostasis to be maintained. Skin aging is induced by internal and external factors. The loss of moisture represents an internal factor of skin aging and follows a natural course through a reduction in the regulation of hyaluronan synthesis or the loss of keratinocyte tight junctions [57,58,59]. Here, we have identified factors that increase moisturization, including occludin, which is a regulator of tight junctions, and HAS-1, which regulates hyaluronan synthase. Vitamin E is used heavily worldwide to protect cells from oxidative stress and aging [60] and is an essential nutrient and a powerful antioxidant. It is a fat-soluble vitamin that occurs naturally in eight forms. Vitamin E can be divided into two principal classes: tocopherols and tocotrienols. These can be further categorized into slightly different compounds, known as alpha, beta, delta, and gamma [61]. Despite its widespread use as a lipid-soluble antioxidant, few studies have examined the moisturizing effects of vitamin E on the skin. Therefore, we also determined the protective activity of vitamin E on skin moisturization.

The evidence indicates that UVB induces acute and chronic skin problems, such as dehydration, and generates reactive oxygen species that progress skin aging [62]. Currently, there is an increased interest in skin health and natural products that prevent photoaging or are involved in skin protection [54]. In fact, clinical researchers are also trying to find new treatments for anti-oxidation and anti-photoaging. In previous reports, flavonoids or phenolics that are abundant in extracts of plants are related to antioxidant ability, and they are even considered an indispensable ingredient in various nutritional, pharmaceutical, and cosmetic applications [63,64,65,66]. In this study, we investigated the potential skin-protective functions of As-EE by evaluating the expression of genes related to antioxidant and moisturizing capacity. As shown in Figure 1, the ability of As-EE to reduce free radical levels in cell-free systems was confirmed using ABTS, DPPH, FRAP, and CUPRAC assays. Based on the IC_50_ values (46 µg/mL and 13 µg/mL) of As-EE for DPPH and ABTS, it is assumed that this plant can have a higher antioxidative activity than other plants such as *Malus baccata*, *Canarium subulatum*, *Licania macrocarpa*, *Atriplex halimus*, and *Euphorbia resinifera*, with IC_50_ values of 50 to 200 μg/mL [67,68,69,70,71,72,73,74,75,76]. The contents of rutin and quercetin were calculated to be 0.53 and 0.03%, respectively, using standard area curves of these compounds (data not shown). These results suggest that the antioxidative property of As-EE can be beneficial as major pharmacological activities and rutin and quercetin can be considered as active components in As-EE.

An emerging study has shown that MAPK was activated in UV-induced signal transduction [77]. We found that As-EE has restorative effects on UVB-induced skin damage by reducing ERK1/2, not just in antioxidant effects. Furthermore, ERK1/2 directly interacts with occludin and can activate TGMs [78,79]. Occludin has a critical role in protecting the skin barrier by maintaining tight junctions in cell–cell junctions from the irradiation of UVB [80]. Our data have demonstrated that As-EE improved skin-hydrating effects by elevating moisturizing factors, including occludin and TGM-1, that were inhibited by UVB. After HaCaT cells were treated with As-EE, we found that the expression of occludin and TGM-1 was enhanced in a dose-dependent manner. In addition, As-EE can dramatically restore the levels of occludin and TGM-1 under UVB irradiation, showing that As-EE can recover skin water loss caused by UVB. These results suggest that As-EE might be a feasible anti-aging ingredient in future cosmetics.

Moreover, previous research has demonstrated that oxidative stress also induces apoptotic cell death [81]. Owing to As-EE antioxidant effects in UVB-exposed human keratinocytes, we determined the anti-apoptotic ability of As-EE in the same cell line. We analyzed cell death using propidium iodide (PI)-Annexin V staining. As expected, As-EE markedly reduced the level of apoptosis in UVB-irradiated HaCaT cells. These results forcefully suggest that As-EE has a protective activity against UVB-induced apoptotic death in human keratinocytes. These results suggest that As-EE can be a feasible anti-aging ingredient in cosmetics.

Which compounds can mediate the anti-photoaging activity of As-EE was not elucidated yet. However, based on previously published papers, rutin seems to work as a major active ingredient in As-EE. Numerous papers regarding the role of rutin in photoaging effects have been published so far. For example, rutin was reported to be cytoprotective against oxidative stress in skin fibroblasts and cytotoxic conditions under UVB exposure [82,83]. Solid lipid nanoparticles containing rutin showed an efficient protective activity against UVB-induced cell death, lipid peroxidation, and metalloproteinase formation [84]. Similar photoprotective activities of apple, *Satureja hortensis*, and *Aronia melanocarpa* were also found to be mediated by its ingredient rutin [85,86,87]. Therefore, these reports raise a possibility that the anti-photoaging activity of As-EE could be in part mediated by rutin. To verify this possibility, further study will be continued with rutin.

## 4. Materials and Methods

### 4.1. Materials

Ethanol extract [70% (w/v)] of the leaves of *A. silvestris* (Ac-EE) was obtained from the National Institute of Biological Resources (Ministry of Environment, Incheon, Korea). Gallic acid, anhydrous sodium acetate, glacial acetic acid, Folin and Ciocalteu’s phenol reagent, 1,1-diphenyl-2 picrylhydrazyl radical (DPPH), ethanol, L-ascorbic acid, 2,20-azino-bis (3-ethylbenzothiazoline-6-sulfonic acid) diammonium salt (ABTS), potassium persulfate, acetic acid buffer, CuCl_2_·2H_2_O, NH_4_Ac, neocuproine, 2,4,6-tri(2-pyridyl)-s-triazine (TPTZ), FeCl_3_·6H_2_O, dimethyl sulfoxide (DMSO), trolox, LiChrosolv^®^ water for chromatography (LC-MS Grade), DL-α-tocopherol acetate, the four inhibitors [SB203580 (p38 inhibitor), SP600125 (JNK inhibitor), U0126 (ERK inhibitor) and Bay117082 (inhibitor of κB kinase)], polyethylenimine (PEI), and bovine serum albumin (BSA), were obtained from Sigma (St. Louis, MO, USA). (3-4,5-dimethylthiazol-2-yl)-2,5-diphenyl-tetrazolium bromide (MTT) was obtained from Amresco (Brisbane, Australia). HaCaT cells (human keratinocyte cell line) and HEK293T (human embryonic kidney cell line) cells were purchased from the American Type Culture Collection (Rockville, MD, USA). Dulbecco’s Modified Eagle’s Medium (DMEM), antibiotics (Penicillin–Streptomycin Solution), and trypsin 0.25% (1X) solution were purchased from Cytiva HyClone (Logan, UT, USA). Fetal bovine serum (FBS) was purchased from Gibco (Grand Island, NY, USA). The 1X phosphate buffered saline (PBS) was procured from Samchun Pure Chemical Co. (Gyeonggi-do, Korea). TRIzol reagent was bought from Molecular Research Center, Inc. (Cincinnati, OH, USA). The cDNA synthesis kits were obtained from Thermo Fisher Scientific (Waltham, MA, USA). The primers for polymerase chain reaction (PCR) were synthesized by Macrogen (Seoul, Korea) and reverse transcription polymerase chain reaction (RT-PCR) premix was purchased from Bop-D Inc. (Seoul, Korea). The luciferase reporter assay system kit was bought from Promega (Madison, WI, USA). The 3 MM CHR was bought from Whatman GE Healthcare Life Sciences. Polyvinylidene fluoride (PVDF) membranes were purchased from Merck Millipore (Burlington, MA, USA) and the Western blot detection kit was bought from ATTO CORPORATION. Some total and phosphor-forms of antibodies for Western blotting specific for each target protein were purchased from either Cell Signaling Technology (Beverly, MA, USA) or Santa Cruz Biotechnology (Santa Cruz, CA, USA). The UVB lamp Bio-link crosslinker BLX-312 was purchased from Vilber Lourmat, Collegien, France. The FITC-Annexin V Apoptosis Detection Kit I was obtained from BD Bioscience (San Jose, CA, USA).

### 4.2. Prepartion of As-EE and HPLC Analysis

The leaves of *A. silvestris* identified by Dr. Van Dung Luong (Dalat University, Vietnam) were collected from Vietnam on 30 August 2016. A voucher specimen (#501) was deposited in the herbarium of the National Institute of Biological Resources. The dried leaves of *A. silvestris* were pulverized and extracted with 70% ethanol at room temperature. The ethanol could completely evaporate because the extract was filtered and concentrated in vacuo at 40 °C. The leftover aqueous solution was vacuum-concentrated and freeze-dried [88]. The phytochemical characteristics of As-EE were determined with HPLC as before [89]. For analysis, a system equipped with a KNAUER (Wellchrom) HPLC-pump K-1001, a Wellchrom high-speed scanning spectrophotometer K-2600, a four-channel deaerator K-500, and a Phenomenex Gemini C18 ODS (5 µm) column was used [90,91,92]. Solvent A (0.1% H_3_PO_4_ in H_2_O) and solvent B (acetonitrile) were used as elution solvents. Rutin, quercetin, hesperidin, and kaempferol were used as standard compounds for HPLC (Figure 1e–m).

### 4.3. Determination of Total Phenolic Content

The total phenolic content (TPC) of the As-EE was measured using Folin and Ciocalteu’s phenol (FC) reagent according to the method of Ali Ghasemzadeh et al. with some modification [93]. A 100 µL volume of As-EE (0–200 µg/mL, previously prepared) dissolved in DMSO or gallic acid (0–500 µg/mL) was dissolved in distilled or deionized water. Then 300 µL of distilled or deionized water and 100 µL of 10% (v/v) Folin and Ciocalteu’s phenol (FC) reagent were added in E-tube. After 5 min of incubation at room temperature, 500 µL of distilled or deionized water and 500 µL of 8% (w/v) sodium carbonate were mixed. After 30 min of incubation at room temperature, using a spectrophotometer, the absorbance of each fraction was measured at 765 nm after 200 µL of the mixture had been poured into a 96-well plate (BioTek Instruments Inc., Winooski, VT, USA). The total phenolic content (TPC) is given as mg of gallic acid equivalent/g of As-EE, and in this method, gallic acid was used as a reference standard (y = 0.0032x + 0.0481, R^2^ = 0.999).

### 4.4. DPPH Assay

DPPH is a method for predicting antioxidant activity. The DPPH-radical-scavenging ability can be used to identify the free-radical-scavenging ability. To determine the oxidant-scavenging capacity of As-EE, a DPPH decolorimetric assay was conducted [94]. First, DPPH (Sigma, St. Louis, MO, USA) was dissolved in methanol and configured into 3 mM stocks, L-ascorbic acid (100 mM) was dissolved in PBS (Samchun Pure Chemical Co. Gyeonggi-do, Korea), and As-EE (100 mg/mL) was dissolved in DMSO separately [95]. Next, the DPPH stock solution was diluted with methanol to 250 µM, L-ascorbic acid was diluted to 50 µM, and As-EE was serial diluted from 200 µg/mL to 0 µg/mL. These mixtures were incubated with foil at 37 °C for 30 min and then the absorbance was measured at 517 nm using a spectrophotometer (BioTek Instruments Inc., Winooski, VT, USA). The percentage of inhibition for DPPH scavenging was estimated as follows: DPPH scavenging effect (%) = [(A_0_ − A_1_)/A_0_] × 100 in which A_0_ is the absorbance of DPPH and A_1_ is the absorbance of the sample (As-EE or L-ascorbic acid).

### 4.5. ABTS Assay

Another technique for assessing antioxidant-scavenging properties is the ABTS-radical-scavenging assay. First, ABTS and potassium persulfate (K_2_S_2_O_8_) were taken using a chemical balance. ABTS was dissolved in PBS and potassium persulfate was dissolved in acetic acid buffer solution. Then, 7.4 mM ABTS and mixed with 2.4mM potassium persulfate in a ratio of 1:1. After incubating the solution for 30 min at 37 °C in the dark, we checked if the solution color changed to dark green. In a 96-well plate, the ABTS solution and As-EE (0–200 µg/mL) were mixed at a 1:1 ratio. A positive control was utilized, which was L-ascorbic acid (50 µM). The mixture was covered with foil and incubated once again at 37 °C for 30 min in an incubator (Thermo Fisher Scientific, Waltham, MA, USA) [76]. Using a spectrophotometer, the absorbance of each fraction was measured at 730 nm after 30 min of incubation at 37 °C (BioTek Instruments Inc., Winooski, VT, USA). The following percentage was computed for the ABTS0-scavenging effect: Assuming that A0 is the absorbance of ABTS and A1 is the absorbance of the sample, the ABTS-scavenging effect (%) is calculated as [(A_0_ − A_1_)/A_0_] × 100. (As-EE or L-ascorbic acid).

### 4.6. Cupric Ion Reducing Antioxidant Capacity (CUPRAC) Assay

The CUPRAC assay is a redox reduction between the CUPRAC reagent and the antioxidants with a leading thiol group (for example, glutathione) present in the sample. In this process, the reagent reduces itself, forming a chelate complex of copper (I)-neocuproine, which provides a color measurable at 450 nm [96]. First, CuCl_2_⋅2H_2_O was dissolved with distilled or deionized water to make a copper (II) chloride solution at a concentration of 10 mM. Ammonium acetate (NH_4_Ac) was taken using a chemical balance and dissolved in distilled or deionized water to prepare NH_4_Ac buffer at pH 7.0. Neocuproine (Nc) was dissolved in pure EtOH to make a neocuproine solution at a concentration of 7.5 mM. Then, the copper(II) chloride solution, neocuproine solution, and NH_4_Ac buffer were mixed in a 15 mL conical tube at a ratio of 1:1:1 (v/v/v). Next, 600 µL of the mixture and 200 µL of As-EE (0–200 µg/mL, previously prepared) were put into 1.5 mL E-tubes. A 200 µL volume of the reaction solution was added into a 96-well plate. Trolox (0.4 mM) was used as a positive control. After incubation for 1 h, a spectrophotometer was used to measure the absorbance of each fraction at 450 nm (BioTek Instruments Inc., Winooski, VT, USA).

### 4.7. Ferric-Reducing Antioxidant Power (FRAP) Assay

First, anhydrous sodium acetate and glacial acetic acid were taken using a chemical balance and dissolved with distilled or deionized water to make acetate acid buffer (pH 3.6) at a concentration of 300 mM. TPTZ was dissolved in distilled or deionized water, and concentrated hydrochloric acid was added to make a TPTZ solution at a concentration of 10 mM. FeCl_3_·6H_2_O was dissolved with distilled or deionized water to make a FeCl_3_ solution at a concentration of 20 mM. Then, the acetic acid buffer, TPTZ solution, and FeCl_3_ solution were mixed according to the ratio 10:1:1 (v/v/v). A 100 µL volume of the As-EE solution (0–200 µg/mL) was added. In a 96-well plate, 100 µL of FRAP working solution was added and the plate was shaken well. Additionally employed as a positive control was trolox (0.4 mM). Using a spectrophotometer, the absorbance of each fraction was measured at 593 nm after 15 min of incubation in the dark at 37 °C (BioTek Instruments Inc., Winooski, VT, USA).

### 4.8. Cell Culture

HaCaT cells (human keratinocyte cell line) were cultured in DMEM supplemented with 1% (v/v) penicillin–streptomycin and 10% (v/v) FBS. HEK293T (human embryonic kidney cell line) cells were cultured in DMEM with 1% (v/v) penicillin–streptomycin and 5% v/v FBS. All cells were grown in an incubator with 5% CO2 humidity (Thermo Fisher Scientific, Waltham, MA, USA) at 37 °C. To maintain fresh cells, the cells were divided and given fresh medium three times per week.

### 4.9. Cell Viability Test

To appraise the cytotoxicity of As-EE, HaCaT cells were seeded into 96-well plates at 5 × 10^5^ cell/mL in DMEM supplemented with 1% (v/v) penicillin–streptomycin and 10% (v/v) FBS overnight. As-EE was applied to all cells in a dose-dependent manner for 24 h. To evaluate the cytotoxicity of As-EE, HaCaT cells were seeded into 24-well plates at 3 × 10^5^ cell/mL in DMEM overnight. After pre-treating with As-EE for 30 min, the media was sucked out, and the cells were washed with PBS. Then, the cells were covered with PBS and irradiated with 30 mJ/cm^2^ UVB before the PBS was sucked out again and As-EE was added in a dose-dependent manner, with a subsequent 24 h incubation. Following the removal of 100 µL, 10 µL of MTT solution (5 mg/mL) was put into each well for 3–4 h. MTT stopping solution [10% (w/v) sodium dodecyl sulfate in 1 M HCl] was added when purple formazan appeared to stop the reaction [97]. After incubation for 16–20 h, using a microplate reader, the absorbance of each well was determined at 570 nm (BioTek Instruments Inc., Winooski, VT, USA).

### 4.10. mRNA Preparation and Reverse Transcription Polymerase Chain Reaction (PCR)

To ascertain the gene expression of occludin, HAS-3, TGM-1, HAS-1, claudin, HAS-2, and HAS-3, HaCaT cells were seeded in 6-well plates at a density of 5 × 10^5^ cells/mL. The total RNA was isolated using TRIzol reagent, according to the operating manual. The cDNA synthesis was performed using a cDNA synthesis kit. RT-PCR was implemented using PCRBIO HS Taq PreMix (PCR Biosystems Ltd., Oxford, UK) after preparing cDNA with reverse transcriptase [97]. GAPDH was used as reference gene in RT-PCR. The primers for this study are listed in Table 1.

### 4.11. Reporter Gene Assays

HEK293T cells (1 × 10^5^ cells/well) were plated in a 24-well plate using DMEM supplemented with 5% (v/v) FBS overnight. After removing the cultural supernatant, fresh medium (400 µL) was added into each well. Then, the cells were transfected with 0.8 μg/mL of luciferase construct (eg., CREB-Luc) and β-gal (control) by adding the transfection reagent [polyethylenimine (PEI)]. After 24 h incubation, the media was changed to DMEM supplemented with 1% (v/v) penicillin–streptomycin and 5% (v/v) FBS first. Then cells were treated with 50, 75, and 100 µg/mL of As-EE or 6.25 and 12.5 µM of ascorbic acid, respectively, for a further 24 h. A luminometer was used to measure the luciferase activity (BioTek Instruments Inc., Winooski, VT, USA).

### 4.12. Total Cell Lysate Preparation

First, As-EE-treated HaCaT cells were collected with cold PBS. After adding lysis buffer, cells were incubated for 15 min on ice. The cell lysates were then kept at −70 °C after being centrifuged for 15 min at 12,000 rpm. To obtain protein samples for Western blotting, the protein concentrations were measured at 570 nm using the Bradford protein assay (Bio-Rad, Hercules, CA, USA) as described previously [98,99].

### 4.13. Immunoblotting Analysis

After making protein loading samples, all samples were subjected to SDS-polyacrylamide gel electrophoresis (SDS-PAGE), and transferred onto PVDF membranes (Millipore, Billerica, MA, USA). Every loading sample contained 20 µg of proteins. The membranes were blotted with 3% (w/v) BSA at room temperature for 30 min and incubated with primary antibodies overnight at 4 °C. Following that, TBST was used to wash all membranes three times for a total of ten minutes each time. All membranes were then washed once again, followed by a second 2 h incubation with the secondary antibody at room temperature. Finally, all membranes were detected with chemiluminescence reagents [100,101]. In this method, β-actin was used as an immunoblotting loading control, as reported previously [102].

### 4.14. UVB Irradiation

In a 6-well pate, HaCaT cells were evenly plated at a density of 3 × 10^5^ cells/well and incubated overnight. Before UVB irradiation, As-EE was used to pre-treat cells for 30 min. PBS was used to wash the cells and the 6-well pate was irradiated with 30 mJ/cm^2^ UVB [103]. After UVB irradiation and removing PBS, As-EE was treated again and incubated at 37 °C with 5% CO_2_.

### 4.15. Morphological Changes

To determine morphological changes, HaCaT cells (3 × 10^5^ cells/well) were seeded equably in a 6-well pate using DMEM supplemented with 10% (v/v) FBS and 1% (v/v) penicillin–streptomycin overnight. Cells were pre-treated with As-EE for 30 min and irradiated under UVB light (Bio-Link BLX-312; Vilber Lourmat, Collégien, France) with a strength of 30 mJ/cm^2^, as established previously [104,105,106,107,108,109]. The PBS was removed and the cells were treated again with As-EE for 6 h and 12 h [110]. Cells images were taken using an epifluorescence microscope (Olympus, Tokyo, Japan).

### 4.16. PI and Annexin V Staining (FACS)

To evaluate apoptosis, HaCaT cells were seeded into 6-well plates and incubated overnight. The cells were pre-treated with As-EE (from 0 µg/mL to 100 µg/mL) for 30 min. Cells were then removed and washed with PBS. The washed cells were irradiated with UVB (30 mJ/cm^2^). After incubation for 24 h, UVB-irradiated cells were harvested, washed with cold PBS, and centrifuged at a speed of 1200 rpm for 3 min at 4 °C. Next, all samples were prepared using an FITC-Annexin V Apoptosis Detection Kit I (BD Bioscience, San Jose, CA, USA). A 100 µL volume of 1 × binding buffer was added first. All cells were stained with two apoptotic markers (FITC and PI) following the manufacturer’s instructions [67]. After incubation, 400 µL of 1 × binding buffer was added and the fluorescence was measured using a Guava easy Cyte flow cytometer (Millipore, Burlington, MA, USA).

### 4.17. Statistical Analysis

All results presented are expressed as the mean ± standard deviation (SD) of experiments performed with six (Figure 2a and Figure 3a,e) or three samples (Figure 1a–d, Figure 2e and Figure 4b,c,e,g,i,k). IC_50_ values were determined using Graphpad Prism 7.0. The graphs were drawn in this study using SigmaPlot (Systat Software, San Jose, CA, USA). Western blot data are a representative of three. All data were analyzed using Mann–Whitney U tests. *p*-values less than 0.05 and less than 0.01 were regarded as statistically significant and very statistically significant, respectively. Similar experimental data were also observed using an additional independent set of in vitro experiments conducted using the same numbers of samples.

## 5. Conclusions

To sum up, we demonstrated that As-EE may be a potential drug to prevent UVB-induced skin photoaging because we did not see cytotoxicity in the results. At the same time, As-EE also has the ability to moisturize, as summarized in Figure 5. However, we need to study As-EE further in the future—for instance, whether it is difficult to collect As-EE and how stable it is. Moreover, here we did not further investigate inflammation and whitening effects; these functions also need to be explored further. In the end our results strongly suggest that As-EE has antioxidant and moisturizing capacities and that this fraction has potential applications in cosmeceutical preparations. Since GC-Mass analysis can only detect highly gas-flammable compounds, we will also employ LC-Mass analysis for identify the other compounds in the following project.

## Figures and Tables

**Figure 1 plants-12-01167-f001:**
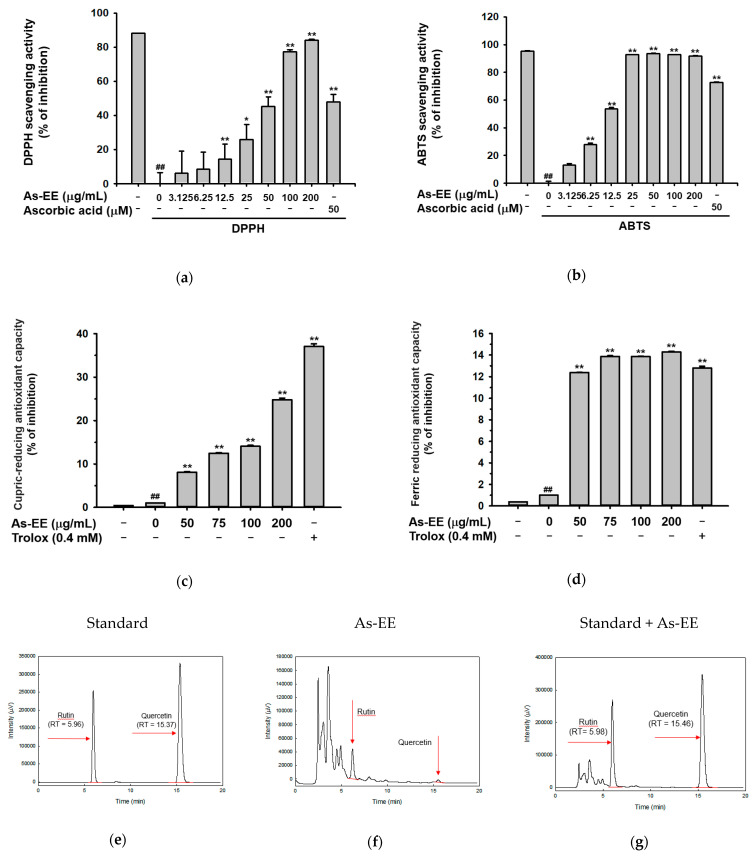
Effects of As-EE on antioxidative capacity. (**a**) As-EE (0 to 200 µg/mL) was incubated with DPPH (250 µM) in foil at 37 °C for 30 min and then the absorbance was measured at 517 nm. Ascorbic acid (50 µM) together with positive controls; (**b**) ABTS and potassium persulfate solution mixed with As-EE (0 to 200 µg/mL) incubated for 30 min in the dark at 37 °C. At 730 nm, the solution’s absorbance was found; (**c**) Copper(II) chloride solution, neocuproine solution, and NH_4_Ac buffer were mixed with As-EE (0 to 200 µg/mL). Trolox (0.4 mM) was used as a positive control. After 1 h of incubation, the absorbance was measured at 450 nm using a spectrophotometer; (**d**) Acetic acid buffer, TPTZ solution, and FeCl_3_ solution were mixed with As-EE (0 to 200 µg/mL). A positive control was employed, which was trolox (0.4 mM). Each fraction’s absorbance was measured at 593 nm after 15 min of dark incubation at 37 °C; (**e**–**m**) The phytochemical profiles of rutin, quercetin, hesperidin, and kaempferol in As-EE were analyzed using HPLC. Results (**a**–**d**) are expressed as the mean ± standard deviation. ^#^
*p* < 0.05 and ^##^
*p* < 0.01 compared with the normal groups. * *p* < 0.05 and ** *p* < 0.01 compared with the control groups.

**Figure 2 plants-12-01167-f002:**
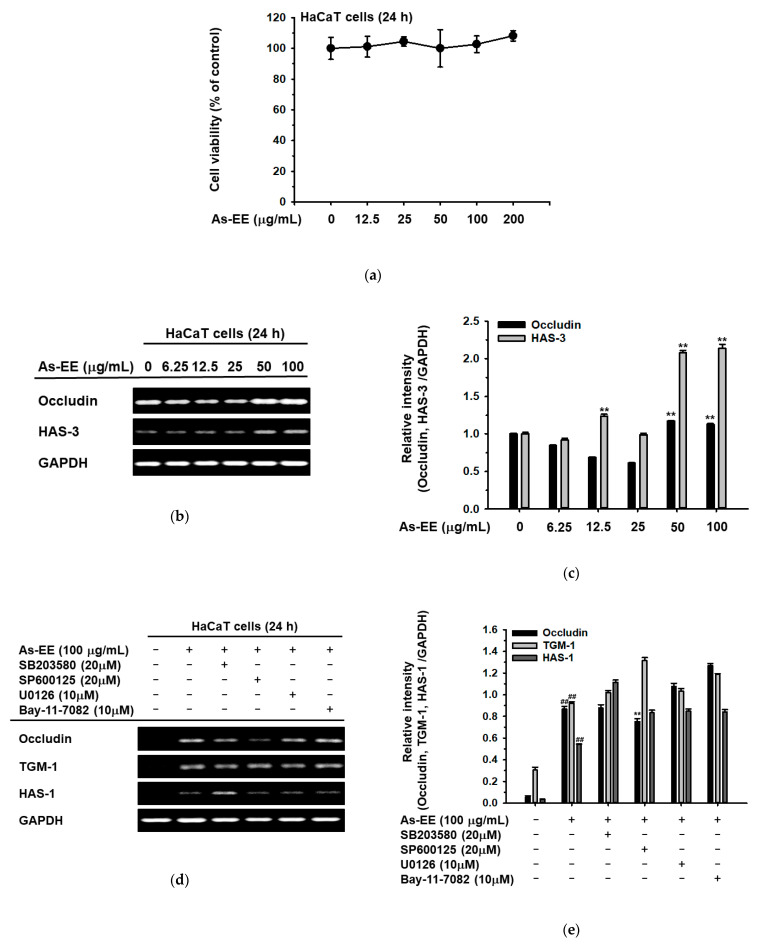
Effects of As-EE on cell viability and skin moisture-protection activity. (**a**) Cytotoxicity of As-EE was measured using a 3-(4-5-dimethylthiazol-2-yl)-2,5-diphenyltetrazolium bromide (MTT) assay in HaCaT cells; (**b**) The expression levels of skin-barrier-related and hydration factors were determined with RT-PCR in HaCaT cells after treatment with As-EE at doses ranging from 0 to 100 µg/mL for 24 h; (**c**) The relative intensity of RT-PCR results (occludin and HAS-3) was measured with ImageJ; (**d**) Inhibitors were applied to HaCaT cells (SB20580, a p38 inhibitor; SP600125, a JNK inhibitor; U0126, an ERK inhibitor; and Bay117082, a κB kinase inhibitor) for 24 h and the mRNA levels of occludin, TGM-1, and HAS-1 were measured with RT-PCR. (**e**) The relative intensity of mRNA levels was measured with ImageJ. ^#^ *p* < 0.05 and ^##^
*p* < 0.01 compared with the normal groups. * *p* < 0.05 and ** *p* < 0.01 compared with the control groups.

**Figure 3 plants-12-01167-f003:**
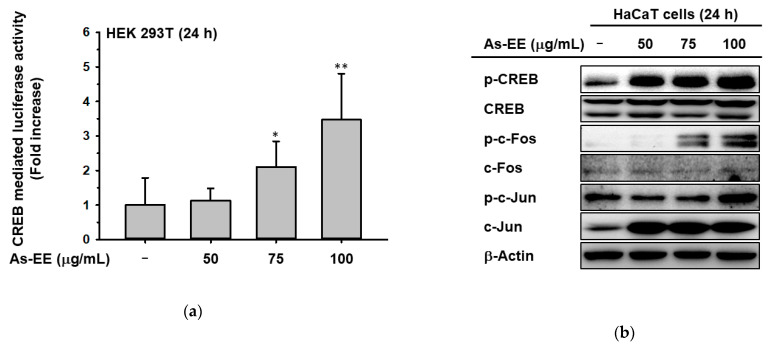
Effects of As-EE on MAPK-mediated AP-1 and CREB signaling pathway. (**a**) HEK293T cells transfected with CREB-Luc were incubated with As-EE for 24 h. A galactosidase construct was used as a control, and luciferase activity was measured using a luminometer; (**b**,**c**) HaCaT cells were incubated with As-EE for 24 h. Immunoblot analysis was used to evaluate the phosphorylation levels of c-Jun, c-Fos, p-CREB, and JNK; (**d**) RT-PCR was used to evaluate the expression levels of skin-barrier-related and hydration factors in HaCaT cells after treatment with vitamin E in a dose-dependent manner from 0 to 100 µM for 24 h; (**e**) Luciferase assay; (**f**) Phosphorylation level of JNK was measured with immunoblotting in HaCaT cells treated with vitamin E for 24 h. * *p* < 0.05 and ** *p* < 0.01 compared with the normal groups.

**Figure 4 plants-12-01167-f004:**
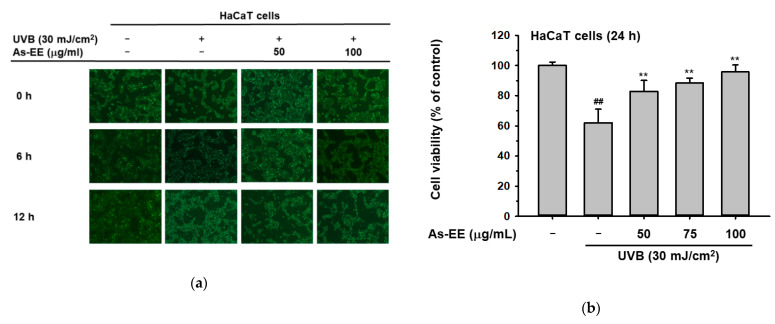
Moisturizing and anti-apoptotic effect of As-EE in UVB-irradiated human keratinocytes. (**a**) The morphology of HaCaT cells with As-EE treatment (50 and 100 µg/mL) under UVB irradiation for 6 and 12 h was examined using microscopy; (**b**) Viability of HaCaT cells was evaluated with an MTT assay in As-EE-treated cells exposed to UVB irradiation; (**c**) FACS analysis in HaCaT cells treated with As-EE under UVB irradiation; (**d**) HaCaT cells were pre-treated with As-EE for 30 min and irradiated with UVB. After incubation for 6 h, the mRNA levels were measured with RT-PCR; (**e**) The relative intensity of mRNA was measured with ImageJ; (**f**) Phosphorylation levels of ERK were checked with an immunoblot analysis; (**g**) The relative intensity of protein was measured using ImageJ; (**h**) Phosphorylation levels of CREB were checked with an immunoblot analysis; (**i**) The relative intensity of protein was measured using ImageJ; (**j**) HaCaT cells were treated with MAPK inhibitors (SB20580, a p38 inhibitor; SP600125, a JNK inhibitor; and U0126, an ERK inhibitor) for 6 h under UVB irradiation and mRNA levels were determined with RT-PCR; (**k**) The relative intensity of mRNA levels was measured with ImageJ. * *p* < 0.05 and ** *p* < 0.01 compared with the control groups (only UVB group). ^#^
*p* < 0.05 and ^##^
*p* < 0.01 compared with the normal groups.

**Figure 5 plants-12-01167-f005:**
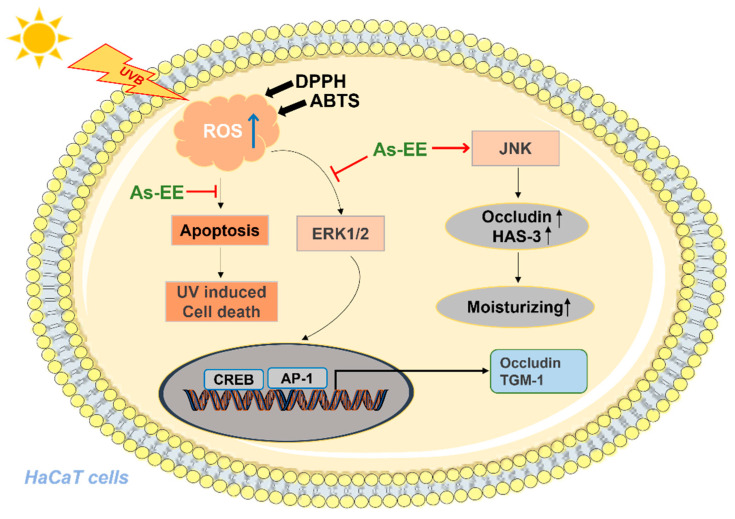
Summary of pathways regulated by As-EE related to its moisturizing and anti-photoaging effects.

**Table 1 plants-12-01167-t001:** List of primers synthesized for RT-PCR.

PCR Type	Gene Name	Sequence (5′-3′)	
RT-PCR	Occludin	Forward	GAAGATGAGGATGGCTGTCA
		Reverse	AAATTCGTACCTGGCATTGA
	HAS-3	Forward	TATACCGCGCGCTCCAA
		Reverse	GCCACTCCCGGAAGTAAGACT
	TGM-1	Forward	GAAATGCGGCAGATGACGAC
		Reverse	AACTCCCCAGCGTCTGATTG
	HAS-1	Forward	CCACCCAGTACAGCGTCAAC
		Reverse	CATGGTGCTTCTGTCGCTCT
	Claudin	Forward	AGGAACACATTTATGATGAGCAG
		Reverse	GAAGTCATCCACAGGCGAA
	HAS-2	Forward	TTCTTTATGTGACTCATCTGTCTCACCGG
		Reverse	ATTGTTGGCTACCAGTTTATCCAAACG
	GAPDH	Forward	GACAGTCAGCCGCATCTTCT
		Reverse	GCGCCAATACGACCAAATC

## Data Availability

Not applicable.

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
