# Peer review of "Antiphotoaging and Skin-Protective Activities of *Ardisia silvestris* Ethanol Extract in Human Keratinocytes"

_plants, 2023, doi:10.3390/plants12051167_

Round 1

Reviewer 1 Report (Previous Reviewer 4)

Responses are adequate.

Author Response

Thanks very much for your reviewing.

Reviewer 2 Report (Previous Reviewer 3)

In accordance with the reviewers' suggestions, the changes made by the authors   bring clarifications and improve the quality of the manuscript.

Author Response

Thanks very much for your reviewing.

Reviewer 3 Report (Previous Reviewer 1)

In current form (with all changes that have been made) I recommend for publication.

Author Response

Thanks very much for your reviewing.

This manuscript is a resubmission of an earlier submission. The following is a list of the peer review reports and author responses from that submission.

Round 1

Reviewer 1 Report

1 / keywords cannot coincide with the title; please add more

2 / introduction - too short, please expand it

3 / GC-MS chromatogram - illegible, please insert in a more aesthetic version

4 / materials - please add "chemicals and reagents" and provide all used during the study

5 / please provide all devices used during the tests according to the record: name, model (manufacturer, country, city)

6 / please provide in detail all methodologies

7 / for concentrations % - please specify (v / v), (w / w), etc.

8 / for solutions - please give what's dissolved

9 / for determinations in which a standard curve was used - please specify the concentration range of the analyte, the curve equation, its fitting

Author Response

-Reviewer 1

This study is well-prepared and designed one and seems to be enough to be published in this journal. Only minor points were raised to improve the quality of this paper.

  1. Keywords cannot coincide with the title; please add more

Answer: Thanks for your comment. As suggested, all keywords were changed. From “Ardisia silvestris ethanol extract; anti-photoaging; moisturizing;” to “Ardisia silvestris; irradiation; ROS; AP-1; anti-apoptosis; antioxidative capacity”.

  1. Introduction - too short, please expand it

Answer: Thanks for your comment. In introduction part, we have added some additional paragraphs to expand the Introduction section (Please see line 80 to line 96 and line 104 to line 112).

  1. GC-MS chromatogram - illegible, please insert in a more aesthetic version

Answer: Thanks for your comment. As you know, since the GC-mass instrument has its own printing-out format as default, we are not able to adjust style of GC-MS chromatogram. Therefore, we should keep current data. We apologize for this.

  1. Materials - please add "chemicals and reagents" and provide all used during the study

Answer: Thanks for your comment. We have added chemicals and reagents: “Ethanol, acetic acid buffer, anhydrous sodium acetate, glacial acetic acid, FeCl3·6H2O” and we also corrected spelling errors from “FeCl3·6H2O” to “FeCl3·6H2O” in the materials part (see Line 319-321)

  1. Please provide all devices used during the tests according to the record: name, model (manufacturer, country, city)

Answer: Thanks for your comment. We have added necessary information as follows: 1] spectrophotometer (BioTek Instruments Inc., Winooski, VT, USA): Line 369, 396-397, 414, and 426] luminometer (BioTek Instruments Inc., Winooski, VT, USA): Line 466-467] UVB lamp (Bio-Link BLX-312; Vilber Lourmat, Collégien, France): Line 498] FITC-Annexin V Apoptosis Detection Kit I (BD Bioscience, San Jose, CA, USA): Line 508-509.

  1. Please provide in detail all methodologies

Answer: Thanks for your comment. We have added detailed explanation of all methodologies (see all the highlighted ones in 4. Materials and methods section Lines 313-523).

  1. For concentrations % - please specify (v / v), (w / w), etc.

Answer: Thanks for your comment. We have corrected “%” to “% v/v, % w/v, % w/w” as follows:

Line 366: 8% w/v sodium carbonate

Line 364: 10% v/v Folin and Ciocalteu’s phenol (FC) reagent

Line 410: 1:1:1 v/v/v

Line 422: 10: 1: 1 v/v/v

Line: 429-431, 435-436, 490: 1% v/v penicillin-streptomycin and 10% v/v FBS

Line 444: 10% w/v sodium dodecyl sulfate

Line 460: 5% v/v FBS overnight

Line 464: DMEM supplemented with 1% v/v penicillin-streptomycin and 5% v/v FBS

Line 471: 2% v/v NP-40

Line 482: 3% w/v BSA

  1. For solutions - please give what's dissolved

Answer: Thanks for your comment. We have corrected as follows.

Line 362-363: “100 µL of As-EE (0–200 µg/mL, previously prepared) dissolved in DMSO or gallic acid (0–500 µg/mL) dissolved in distilled or deionized water”

Line 377-378: “L-Ascorbic Acid (100 mM) was dissolved in 1X phosphate buffered saline (PBS) (Samchun Pure Chemical Co. Gyeonggi-do, Korea), and As-EE (100 mg/mL) was dissolved in DMSO separately.”

Line 389-390: “ABTS was dissolved in PBS and potassium persulfate dissolved in acetic acid buffer solution.”

  1. For determinations in which a standard curve was used - please specify the concentration range of the analyte, the curve equation, its fitting

Answer: Thanks for your comment. We have corrected as follows.

Line 363: gallic acid (0–500 µg/mL)

Line 369-371: The total phenolic content (TPC) was written as mg of gallic acid equivalent/ g of As-EE, and in this method, gallic acid was used as a reference standard (y = 0.0032x + 0.0481, R2 = 0.999).

Reviewer 2 Report

At a first glance, I found the paper  "Antiphotoaging and skin-protective activities of Ardisia silvestris Ethanol Extract in Human Keratinocytes" interesting and well written.

However, a more careful reading revealed that there are some serious flaws.

Considering this is a submission to Plants, I would like to see a little more information about the species. The whole introduction is about skin and skin protection without a word about the plant, its connection to skin treatment in traditional medicine etc.

Table 1 is simply meaningless. It makes no sense at all to present the phytochemical analysis of an ethanolic extract and state that the major compound  (97,5%) is DMSO, plus carbon dioxide, ethanol, acetic acid and dimethylsulfone accounting for another 1,4%. The rest of the listed compounds are not natural compounds, not what one would expect from an ethanolic extract. Indeed, they seem to the products of a column bleeding. The chromatogram, which the part e) of figure (totally out of place!) confirms that. Also, RT as retention time is not acceptable in analytical chemistry.

Table 2 is useless, just on line, can de described in the text.

So, it seems that this extract is not a plant extract at all, but column bleeding compounds dissolved in DMSO. The whole skin protection evaluation, albeit well designed, falls to nothing. What a pity.

Author Response

-Reviewer 2

At a first glance, I found the paper "Antiphotoaging and skin-protective activities of Ardisia silvestris Ethanol Extract in Human Keratinocytes" interesting and well written.

However, a more careful reading revealed that there are some serious flaws.

Considering this is a submission to Plants, I would like to see a little more information about the species. The whole introduction is about skin and skin protection without a word about the plant, its connection to skin treatment in traditional medicine etc.

Answer: Thanks for your comment. We have added more information about As-EE (see Line 348-354)

Table 1 is simply meaningless. It makes no sense at all to present the phytochemical analysis of an ethanolic extract and state that the major compound (97,5%) is DMSO, plus carbon dioxide, ethanol, acetic acid and dimethylsulfone accounting for another 1,4%. The rest of the listed compounds are not natural compounds, not what one would expect from an ethanolic extract. Indeed, they seem to the products of a column bleeding. The chromatogram, which the part e) of figure (totally out of place!) confirms that. Also, RT as retention time is not acceptable in analytical chemistry.

Answer: Thanks for your comment. Regarding GC-MS chromatogram and Table1, I’m very sorry about this part. Because of the system default we could not change the profile of this. Although some compounds are derived from solvents or background, however, still, others are known to be active or major compounds. Aim of this analysis is to get fingerprinting information of As-EE as well as indicator/standardization compound for its industrial application. Since we have recently fixed GC-MS analysis for fingerprint of plant extracts published in our lab, we have also used GC-MS for this extract. As soon as we collect more fingerpring profiles of plant extracts, we will make standardization protocol for preparing standardization of plant extract for industrial application. Of course, when we decide an indicator compound in this extract, then, we will establish HPLC analysis method.

Table 2 is useless, just on line, can de described in the text.

Answer: Thanks for your comment. We have deleted the Table 2 and described in the text.

So, it seems that this extract is not a plant extract at all, but column bleeding compounds dissolved in DMSO. The whole skin protection evaluation, albeit well designed, falls to nothing. What a pity.

Answer: Thanks for your comment. As you can see, As-EE showed promising anti-photoaging properties via the regulation of mitogen-activated protein kinase, which could be beneficial in the dermatological and cosmeceutical industry. It has also higher amount of phenolic components. Although GC-Mass data included some background materials, we are sure that most compounds are active and effective ones in inducing moisturizing factors with CREB-activating capacity, which is similarly found by vitamin E treatment, as well as UV-protective activity. Since GC-Mass can only detect highly gas-flammable compounds, we will also use LC-Mass for detecting the other compounds. Relevant sentence has been included in Line 532-534.

Reviewer 3 Report

The aim of this study was to evaluate anti photoaging and skin-protective activities of Ardisia silvestris extract using different in vitro methods.

This research is important and can bring valuable information with practical application. The presented research is well-planned, and the manuscript is generally well organized. There were used an appropriate and modern experimental design.

 Overall, the work is well written, although there are some typing errors to correct during the revision of the work and some methodological data are missing. 

The Introduction provides enough data on the stage knowledge of these issues. but in the introduction, the research design is not presented, with a clear presentation of the steps and methods used for answering the research question. In introduction the authors do not clearly state the originality and novelty of this research. What this research brings new regarding this subject? Which are the practical implications of this research?

 It used an appropriate and modern methodology, but more clarifying experimental data are needed. The chemical analysis of the extract is incomplete. Polyphenolic compounds were quantified as total polyphenolic only, and individual polyphenolic compounds and other were not identified and quantified. The nature of the compounds is important for biological effects. They could be identified by an HPLC method, for example. 

For the CUPRAC and FRAP methods - the expression of the results is not clear, they are expressed in which measurement units?

More data related to the vegetal material, identification of the species are needed (who made the identification, where they are stored, the voucher number).

The discussions could take in attention more other literature data related with this subject. and the References could be completed.

Author Response

-Reviewer 3

The aim of this study was to evaluate anti photoaging and skin-protective activities of Ardisia silvestris extract using different in vitro methods. This research is important and can bring valuable information with practical application. The presented research is well-planned, and the manuscript is generally well organized. There were used an appropriate and modern experimental design. Overall, the work is well written, although there are some typing errors to correct during the revision of the work and some methodological data are missing. The Introduction provides enough data on the stage knowledge of these issues. but in the introduction, the research design is not presented, with a clear presentation of the steps and methods used for answering the research question. In introduction the authors do not clearly state the originality and novelty of this research. What this research brings new regarding this subject? Which are the practical implications of this research? It used an appropriate and modern methodology, but more clarifying experimental data are needed.

Answer: Thanks for your comment. According to these comments, we have included some mentions on the originality and novelty of this research (see Line 104-105). We also included potential practical implications of this work (see Line 105-106). Additionally, research design and research methods have been included in Line 108-109. None of previous data regarding this plant in skin has not been published. So, we are unable to include previously published data on this plant in skin. 

The chemical analysis of the extract is incomplete. Polyphenolic compounds were quantified as total polyphenolic only, and individual polyphenolic compounds and other were not identified and quantified. The nature of the compounds is important for biological effects. They could be identified by an HPLC method, for example. 

Answer: Thanks for your comment. This is very good point. This comment is really important when we prepare plant extract to be developed for industrial application. Currently, our data presented in this paper has been supplied for patent application and we have applied an additional project related to development of cosmeceutical preparation with As-EE having anti-aging and anti-photoaging properties. For this, we will develop an analyzing method to determine an indicator compound by HPLC. Since more than 10 compounds were detected by GC-Mass, more time will be needed for setting up the conditions of HPLC and choosing candidate compounds. Therefore, we would like to ask the reviewer the exception of this comment in current version.

For the CUPRAC and FRAP methods - the expression of the results is not clear, they are expressed in which measurement units?

Answer: Thanks for your comment. According to your comment, we have revised the expression of the results clearly (see Line 132-134, 136, Figure 1c and 1d: % of inhibition)

More data related to the vegetal material, identification of the species are needed (who made the identification, where they are stored, the voucher number).

Answer: Thanks for your comment. We have added additional information you requested (see Line 348-354)

The discussions could take in attention more other literature data related with this subject.

Answer: Thanks for your comment. Since available data in both ours and literatures are mostly antioxidative activity values, we have added more sentences mentioning literature data of DPPH and ABTS (see Line 286-291).

and the References could be completed.

Answer: Thanks for your comment. We have revised all references in terms of reference format according to journal’s policy (see References section).

Reviewer 4 Report

Synopsis:

Skin is an important organ which protects the body from environmental damage. UV exposure causes photoaging. Hence protection from photoaging is an important factor in the dermatological and cosmeceutical industry. Ardisia silvestris is a medicinal herb being used around the world. This paper aims to determine the protective effect of As-EE on UV-induced skin aging and cell death as well as its ability to enhance the barrier effect of the skin leading to enhanced hydration. The authors have conducted numerous experiments to support their claims. Their findings imply that As-EE has a protective activity against UVB-induced apoptotic death in human keratinocytes suggesting that As-EE can be a feasible anti-aging ingredient in cosmetics. Additionally, this paper also determines the protective activity of vitamin E on skin moisturization. While these findings are novel and of interest to the field, there are several inconsistencies in this manuscript that need to be clarified. We recommend substantial revisions before resubmission to Plants.

Revisions/comments:

1.        In the Introduction, line 67-74, the authors mention that “MAPK pathways comprise three diverse pathways…” but they only mention JNK and ERK. Likely p38 was intended to be mentioned here as well,  but was left out. Since p38 inhibitors are used later in the paper, this should be corrected.

2.       While some experiments seem to be performed in triplicate and statistically analyzed, others do not. This reviewer was unable to find any statement that findings were reproduced for at least n = 3. If quantification is not shown, it should be clarified that the findings were reproduced in several experiments (see Figure 3 and Figure 4). This is especially notable for Figure 4a, lower panel. Given that the cell viability results seem significant, the apoptosis inhibition by As-EE seems to not be very significant. This should be clarified.

3.       In Figure 4a, left panel, it is difficult for the reader to confirm differences in floating cells between the groups, even when the picture is enlarged. Quantification would help. Also, were these cells stained with a fluorescent dye prior to imaging?

4.       Authors should be careful to not over-interpret the findings that Occludin expression directly proves that skin hydration will be improved. This is a change in a marker studied in vitro. It might be more correct to say that As-EE leads to changes in biomarkers linked to skin hydration. This refers to statements on lines 227 and 266.

5.       The second Figure 4 (schema) should actually be labeled Figure 5.

6.       There are inconsistencies between the data shown in figure 4c, 4e, and Figure 5 (schema). In Figure 4c the authors show what As-EE inhibits UVB-induced pERK, but in figure 4e they show that an inhibitor of MEK/ERK, U0126, restores UV-induced inhibition of Occludin and TGM-1. This is counter to the schema in the last figure.

7.       It is not mentioned in the paper why 30mJ/cm2 of UVB dosage was used. Was a study conducted to determine the dosage?

Author Response

-Reviewer 4

Skin is an important organ which protects the body from environmental damage. UV exposure causes photoaging. Hence protection from photoaging is an important factor in the dermatological and cosmeceutical industry. Ardisia silvestris is a medicinal herb being used around the world. This paper aims to determine the protective effect of As-EE on UV-induced skin aging and cell death as well as its ability to enhance the barrier effect of the skin leading to enhanced hydration. The authors have conducted numerous experiments to support their claims. Their findings imply that As-EE has a protective activity against UVB-induced apoptotic death in human keratinocytes suggesting that As-EE can be a feasible anti-aging ingredient in cosmetics. Additionally, this paper also determines the protective activity of vitamin E on skin moisturization. While these findings are novel and of interest to the field, there are several inconsistencies in this manuscript that need to be clarified. We recommend substantial revisions before resubmission to Plants.

Revisions/comments:

  1. In the Introduction, line 67-74, the authors mention that “MAPK pathways comprise three diverse pathways…” but they only mention JNK and ERK. Likely p38 was intended to be mentioned here as well, but was left out. Since p38 inhibitors are used later in the paper, this should be corrected.

Answer: Thanks for your comment. We have added additional information about p38 (see Line 72-75).

  1. While some experiments seem to be performed in triplicate and statistically analyzed, others do not. This reviewer was unable to find any statement that findings were reproduced for at least n = 3. If quantification is not shown, it should be clarified that the findings were reproduced in several experiments (see Figure 3 and Figure 4). This is especially notable for Figure 4a, lower panel. Given that the cell viability results seem significant, the apoptosis inhibition by As-EE seems to not be very significant. This should be clarified.

Answer: Thanks for your comment. For this issue, we have explained detailed information in Section 4.17 in p24 Line 515-523.

  1. In Figure 4a, left panel, it is difficult for the reader to confirm differences in floating cells between the groups, even when the picture is enlarged. Quantification would help. Also, were these cells stained with a fluorescent dye prior to imaging?

Answer: Thanks for your comment. We feel very sorry for this. We did this experiment for three times, and these results are mean value of these three as mentioned in Statistical analysis section (see L515-523). In addition, no fluorescent dye was used to stain the cells.

  1. Authors should be careful to not over-interpret the findings that Occludin expression directly proves that skin hydration will be improved. This is a change in a marker studied in vitro. It might be more correct to say that As-EE leads to changes in biomarkers linked to skin hydration. This refers to statements on lines 227 and 266.

Answer: Thanks for your comment. We have corrected from “enhance skin moisturizing levels” to “lead to change in biomarkers linked to skin hydration”. Please see Line 256 to 257.

  1. The second Figure 4 (schema) should actually be labeled Figure 5.

Answer: Thanks for your comment. We have corrected from “4” to “5”. Please see Line 527 and 530.

  1. There are inconsistencies between the data shown in figure 4c, 4e, and Figure 5 (schema). In Figure 4c the authors show what As-EE inhibits UVB-induced pERK, but in figure 4e they show that an inhibitor of MEK/ERK, U0126, restores UV-induced inhibition of Occludin and TGM-1. This is counter to the schema in the last figure.

Answer: Thanks for your comment. Our research revealed that As-EE inhibited UVB-induced p-ERK in Figure 4c. As shown in Figure 4e, when we treated inhibitors under UVB exposure, we found Occludin and TGM-1 have been restored, suggested that JNK and ERK signaling pathways played important roles in moisture-retaining capacity (see Line 243-252)

  1. It is not mentioned in the paper why 30mJ/cm2of UVB dosage was used. Was a study conducted to determine the dosage?

Answer: Thanks for your comment. We added references to support it (see Line 498-499).

Round 2

Reviewer 1 Report

I recommend for publication in current form.

Author Response

Thanks for your comment.

Reviewer 2 Report

Unfortunately, the issues raised in the first version are still there. 

There is no way to know the composition of the extracts. The GC-MS table shows only the solvent and the bleeding of the column. 

So what's  the point of this paper? Is DMSO the responsible for the activity? 

Sorry I must reject this paper

Author Response

Answer: Thanks for your comment. We feel very sorry for this part. GC-MS result is a kind of phytochemical fingerprinting data to discriminate this extract to others. This data is helpful this extract to compare other extracts prepared with the same plant in terms of different time (season) and collecting area. We also prepared As-EE using ethanol and ethanol and remaining water have been completely evaporated and dried. So, none of solvent are remained in the extract. Tracers or indentities of some solvent like DMSO are from solvent for GC mass analysis. Additionally, we also used the same amount of DMSO in all experiments, but vehicle control did not show any inhibitory or activating activities. Therefore, we are sure that our effects are not derived by solvent like DMSO.

Reviewer 3 Report

In accordance with the reviewers' suggestions, the changes made by the authors   bring clarifications and improve the quality of the manuscript.

Author Response

Thanks for your comment.

Reviewer 4 Report

Responses are adequate.

Author Response

Thanks for your comment.

Round 3

Reviewer 2 Report

Nothing changed. The recommendation is again Reject